# Prevalence of active trachoma and associated factors among children 1–9 years old at Arsi Negele Town, West Arsi Zone, Oromia Regional State, Southern Ethiopia

Jemal Mekonnen[1], Jeylan Kassim[2], Muluneh Ahmed[3]*, Negeso Gebeyehu[4]

1 Department of Malaria Prevention & Control, West Arsi Zone Health Office, Shashemene, Ethiopia,
2 Department of Public Health, Madda Walabu University Goba Referral Hospital, Bale Goba, Ethiopia,
3 Department of Nursing, School of Health Science, Madda Walabu University, Shashemene campus, Shashemene, Ethiopia, 4 Department of Midwifery, School of Health Science, Madda Walabu University, Shashemene Campus, Shashemene, Ethiopia

* mulunehamed@gmail.com

## Abstract

### Background

Trachoma is a public health issue in more than 50 nations worldwide, mainly in Sub-Saharan Africa, where hundreds of millions of people are considered blind. Ethiopia is projected to have 30% of the global active trachoma burden. The frequency of Trachoma Folliculitis in children aged 1 to 9 years old is 30% in the Oromia Region. Therefore, the aim of this study was to assess the prevalence of active trachoma and associated variables among children aged 1 to 9 years old in Arsi Negele Town, West Arsi Zone, Oromia Regional State, Southern Ethiopia, December 24–26, 2019.

### Methods

A community-based cross-sectional study was conducted in the Arsi Negele town community on December 24–26, 2019. A total of 178 study volunteers were recruited using a single population proportion formula and assigned to families in the town's three kebeles in proportion. A simple random selection procedure was used to choose study participants from the identified households. Madda Walabu University provided ethical approval, and different government structures provided letters of permission. Pre-tested structured questionnaires and binocular loupes X 2.5 were used to collect data from either mothers or fathers of eligible children for eye examination; torches with bottles of alcohol were used to gather data from either mothers or fathers of eligible children for eye examination. For analysis, data was entered into (IBM, SPSS) version 22. To assess factors associated with active trachoma, bivariate and multivariable logistic regressions were used. The crude and adjusted odds ratios with 95% confidence intervals were calculated to investigate the degree of association between the independent variables and active trachoma. Multivariate logistic regression was used to find connections between dependent and independent variables with a $p \leq$ 0.05 confidence levels and a 95% confidence interval.

**Data Availability Statement:** All relevant data are within the paper and its Supporting Information files.

**Funding:** The authors received no specific funding for this work.

**Competing interests:** The authors have declared that no competing interests exist.

## Result

The prevalence of active trachoma was determined to be 21.91% TF among 178 children aged 1 to 9 years. Flies on children's faces (AOR = 3.427; 95 percent CI: 1.432–8.171), unclean children's faces (AOR = 3.99; 95 percent CI: 1.427–11.158), face washing habits (AOR = 3.064; 95 percent CI: 1.273–7.373), and not using soap while face washing (AOR = 4.564; 95 percent CI 1.561–13.342) were found to be statistically significant associated factors with the prevalence of active trachoma.

## Conclusion

The prevalence of active trachoma was found to be relatively high. Face washing practices and the lack of soap use while washing faces were found as associated factors requiring optimal interventions to prevent trachoma infection among children aged 1–9 years in Arsi Negele town.

## Introduction

Trachoma, the most common infectious cause of blindness, is caused by a conjunctival infection with the bacteria Chlamydia trachomatis. Early infection causes redness and irritation with follicles on the tarsal conjunctiva of the eyes; this may match the World Health Organization (WHO) simplified trachoma grading system description of trachoma inflammation–follicular (TF) [1].

The infection spreads easily from person to person by direct contact with nasal and ocular secretions. It is usually transmitted through the sharing of towels, sheets, or clothing, by flies, or simply by human-to-human contact during normal activities [2]. Trachoma affects an estimated 325 million individuals worldwide, with more than 70% living in Sub-Saharan Africa. Ethiopia, Nigeria, and South Sudan are expected to have the largest illness burden in the region [2].

Trachoma is the greatest preventable cause of blindness in underdeveloped nations, mainly in Africa, including Ethiopia. Because of poor living conditions, such as arid rural areas, poor environmental cleanliness, insufficient water supply, and overall poor socioeconomic position, the population in these areas is at risk of blindness [3].

This disease thrives in rural areas where individuals have inadequate access to clean water and sanitation. It is spread through contact with an infected person's eye discharge, contaminated objects such as towels, handkerchiefs, fingers, and, in some circumstances, eye-seeking flies [4].

Trachoma is spread through close personal contact, and it frequently infects entire families and communities; crowded living conditions within the family unit appear to enhance the risk of trachoma [5].

Blinding trachoma is a neglected tropical disease (NTD) that is slated for global abolition as a public health issue by the year 2020. The World Health Organization (WHO) recommends following the SAFE method, which consists of an integrated intervention package of surgery, antibiotics, facial hygiene, and environmental improvement, to accomplish elimination [2, 6].

SAFE stands for the Integrated Package of Measures to Treat, Control, and Prevent New Blinding Cases of Trachoma [7]. WHO adopted the SAFE strategy in 1996 as a result of recent developments in trachoma control, specifically: a standardized surgical procedure for trichiasis, the development of community-based control strategies, new information on trachoma risk factors, and research demonstrating effective treatment of active trachoma with azithromycin [2, 8].

Trachoma is deemed endemic in a district if the prevalence of TF among children aged 1 to 9 years is 10% or higher. Trachomatous trichiasis "unknown to the health system" prevalence of fewer than 1 case per 1000 total population is defined as the elimination of trachoma as (i) the prevalence of Trachomatous inflammation-follicular in children aged 1–9 years is less than 5% in each formerly endemic district [9].

Because NTDs (trachoma) are more prevalent in rural and impoverished populations, they have garnered more attention in recent years [10].

This could imply that the distribution of active trachoma in the town community and its removal tactics as a public health problem have been missed and require further investigation in the current study environment.

## Methods and materials

### Study setting and design

A community-based cross-sectional survey was undertaken in three kebeles of Arsi Negele town, west Arsi zone, Oromia region, Ethiopia. Arsi Negele town was founded in 1936. The town is located 231 kilometers south of Addis Abeba, Ethiopia's capital city. There are three administrative kebeles, as well as seven governmental elementary schools and two secondary institutions.

There is only one health center in town, as well as a rural hospital. The current total population of the town (2020) is 136084, with 66562 males and 69522 females. The number of children aged one to nine years old was estimated to be 2722, or 2% of the total population (GTM, 2013).The total number of households is 28,351. (Report of Arsi Negele town health office, unpublished source, 2017).

### Study participant

The survey included all children aged 1 to 9 years old, as well as the heads of their families who lived in the town. Children aged 1–9 years old from chosen households (HHs) in the study area took part in the study.

### Eligibility

**Inclusion criteria.** The interview included all mothers/caregivers and children aged 1 to 9 years old who lived in designated HHs.

**Exclusion criteria.** All children aged 1–9 years old, as well as all mothers/caregivers from eligible HHs, who were unable to hear or speak in the study area, were barred from participating in the interview. Children who were blind or had a serious medical condition were not allowed to participate in the study.

### Sample size determination

The sample size was calculated using a single population formula, and the proportion (p) was drawn from a study done in Leku town, in the South Nations/Nationality Peoples of Republic (SNNPR) region (12 percent) discovered near the study area, with a 95 percent confidence label. In addition, a 10% increase for non-respondents was added to the computed sample size [11]. As a result, the sample size determined was

$$n = \left[ Z_{\alpha/2} \right]^2 p \left( 1 - p \right) / d^2$$

Where; n = required sample size

$Z_{\alpha/2}$ = Value of the standard normal distribution corresponding to a significant level of alpha (at $\alpha = 0.05$) = 1.96

P = proportion, taking prevalence of active trachoma 12% obtained from a previous community based survey.

By substituting the figures to the above formula, the sample size was:-

$$n = [1.96]^2 \times 0.12 \, [0.88] \, / \, [0.05]^2$$

$$n = 162$$

Taking a 10% non-response rate into account, the final sample size for children 1–9 years old or families was 178. This sample size was evaluated for assurance of optimal sample size utilizing Stat Calculation of EPI Info's unmatched case control calculation of sample size.

**Sampling technique.** A simple random sampling procedure was used to choose family heads and their children from the town. Town health extension workers (HEWs) identified families with at least one kid aged 1 to 9 years and provided the list to data collectors. A lottery approach was used to choose study children from the frame. Because more than one eligible (children ages 1–9 years) kid was encountered in some participating families, only one child was selected and enrolled in the study by the lottery method.

## Study variables

**Dependent variable.** The dependent variable was the prevalence of active trachoma among children aged 1 to 9 years in Arsi Negele.

**Independent variable.** As independent variables, socio-demographic factors of the mother, father, and children (maternal and paternal occupation, marital status, level of education of the father and mother, age of the child, sex of the child, family size and number of children aged 1–9 years old, number of rooms) were included.

**Environmental and household factors.** (Cooking condition, daily water available in, window of cooking room, chimney exhaust of cooking room, availability of latrine, type of latrine, latrine utilization, presence of feces in the house compound, flies in the house, waste disposal sites, availability of livestock, livestock in the house, and overcrowding status);

**Child's behavioral factors.** (Frequency of face washing (decided by asking the mother or caregivers), soap used for washing (determined by asking the mother or caregivers), discharge from the eye, discharge from the nose, facial cleanliness, and the presence of flies on the child's face);

**Knowledge about trachoma.** (Knowledge of the trachoma transmission route and prevention; and Prevention and control of Chlamydia trachomatis infection (frequency of Zithromax prophylaxes dose received, trachoma health education).

## Data collection tool

A systematic and face-to-face interview questionnaire was used to collect data. The questionnaire was primarily modified from previously evaluated related literature. To maintain consistency, the questionnaire was first developed in English, then translated into local languages and retranslated back into English.

Other data collection tools used to examine the children's eyes included three binocular examination loupes (x2.5), three torches with spare batteries and bulbs, three clipboards and pens for three data recorders, and three spray bottles containing alcohol to clean the examiner's hands after each successive examination of the children's eyes in each house.

## Data collection procedures

For data collection, data recorders and graders were used. Data recorders were pre-selected health experts based on their past surveys of similar conditions. After two days of instruction, they were divided into three teams, each of which included a grader and a data recorder. Field-work was scheduled to collect data at three kebeles in the town at the same time. Each family's head of household was interviewed in order to collect socio-demographic information about the families. Following the interview, eligible children's eyes were examined in both directions to differentiate or grade the indications of active trachoma. To obtain data from these young-sters, the World Health Organization's simple classification approach for grading trachoma in community-based surveys was employed.

## Data quality assurance

Data quality was ensured by employing health worker data collectors with survey experience in health-related subjects. Based on their previous experience, integrated eye care workers (IECWs) were chosen as graders. Senior ophthalmic professionals provided two days of training to three data recorders and three graders. Graders were then checked by a senior optometrist at Shashe-mene referral hospital to ensure the correctness of grading for active trachoma. Fortunately, all three graders were fit for validation of trachoma infection or active trachoma diagnosis.

Expert translators translated the questionnaire from English into the local language. One week before data collection, 5% of the same questionnaire was pre-tested on 10 children (6 girls and 4 boys) aged 1–9 years and 10 family heads (7 mothers and 3 fathers). Considering similar socio-demographic characteristics in Shashemene town, As a consequence of the pilot test feed-back, appropriate modifications were made and the questionnaire was changed. The data collec-tors and main investigator kept track of the data collection process and its completeness.

## Data analysis & management

Data were entered into the IBM (SPSS) statistical software version 22.0 for analysis. All study variables were displayed in terms of magnitude and percentage using descriptive statistics. At P ≤0.25 and a 95% confidence level, binary logistic regression was used to identify potentially associated factors with active trachoma. The backward LR approach was then used to find the independent predictor variables at P ≤0.05 with a 95 percent level of confidence and to account for confounding factors. Because the majority of the data were categorical, Pearson correlation was used to see if there was any prediction among the associated or predictor variables. Almost all predictor factors remained steady. Each candidate variable's declaration of association was displayed at P≤0.25 and 95 percent confidence level, and the degree of association was evalu-ated in terms of adjusted odd ratio (AOR) with corresponding confidence interval.

## Operational definitions

**Active trachoma.**　The presence of Trachomatous inflammation follicular (TF) in either of the child's eyes [12].

**A room of a house** - a living space with minimum dimension of $2m^*2m = 4m^2$

**Clean face** - a child who did not have an eye discharge and/or nasal discharge at the time of survey [11].

**Fly density** - household-fly density was determined by examining the presence of flies on children's faces and around the doorways for about half a minute [7].

**Graders** - are experienced (More than 3 years) integrated eye care workers (IECWs) assigned to examine the eyes of children to identify active trachoma.

Prevalence of **active trachoma**- This is the proportion of children in the community who at any one time have active disease trachoma folliculitis of (TF) [12].

**Trachoma awareness:** - define trachoma, a mean of transmissions and a method of prevention [12].

## Result

### Socio-demographic characteristics of the study participants

A total of 178 study participants were approached. 100% of the family heads completed the questionnaire. There were 108 (60.7%) moms and the remainders were fathers with children ages 1 to 9 years. In terms of ethnicity, the majority of respondents 114 (64%t) were Oromo, followed by Amhara, with 29 (16.3%t). According to the religion distribution, the majority of respondents (77 (43.3%) and 61 (34.3%) were orthodox and Muslims, respectively.

Respondents (family heads) said that 46 (25.78%), 39 (21.9%), and 37 (20.8%) were farmers, housewives, and merchants, respectively. In terms of family heads' educational status, 71 (39.9 percent), 60 (33.7%, and 34 (19.1%) were primary school complete, illiterate, and secondary school complete, respectively (Table 1).

### Basic services for studied family related to active trachoma

More than half of the respondents, 92 (51.7%), live in homes with exactly two rooms, while 44 (24.7%) live in homes with fewer than two rooms. More than two-thirds of children's homes (130%) prepared their meals in a kitchen outside the living house, but a minority of respondents (40%) cooked their meals in a kitchen with living houses.

Despite the fact that 110 (61.8%) of total dwellings lack chimney exhaust, nearly half of cooking rooms (87.9%) have windows. Piped water was the primary source of water for 139 (78.1%) of those polled. Water was not available on a daily basis for 56 (31.5%) of the family heads. However, the majority of respondents (52.8%) consumed more than 4 Jerricans (80 liters) every day, and the travel time to obtain water (2 trips) was longer than 60 minutes for more than half of the respondents (94.5%) (Table 2).

### Family sanitation, hygiene and awareness related to trachoma

According to both interview and observation methods, the majority of 164 (92.1%) of the houses had latrines, but more than half of these latrines were not standard latrines for the majority of 118 (66.3%) of the households. The majority of household heads, 131 (73.6%), said that both adults and children used the available latrines on a regular basis.

The proportion of households with open defecation in their complex was 67 (36.7%) of all houses investigated. More than half of those polled (50.6%) said they burned domestically produced trash or waste. The remainder is simply disposed of in an open area (44.7%), disposed of at a farm (31.8%), and buried 12 (6.7%). Although 78 (43.8%t) of respondents own domestic animals, the majority (41.0%) keep these animals in a shelter designed specifically for them (Table 3).

### Basic characteristics of the studied children

The vast majority of the children that participated, 128 (71.9%), were between the ages of 1 and 5 years, with the remainder being between the ages of 6 and 9 years. By default, a nearly equal percentage of male children (52.2%) and female children (47.8%) were tested for active trachoma. The vast majority of the children tested, 77 (43.3%), were either in preschool or too young to attend school. Regarding the school drop status of children, 10 (5.6% of the total

**Table 1. Socio-demographic distribution of family heads of children aged 1–9 years at Arsi Negele Town, West Arsi Zone, Oromia Region, Southern Ethiopia, December, 2019.**

| | Variables (n = 178) | Frequency | Percent (100%) |
|---|---|---|---|
| **Sex** | | | |
| | Male | 70 | 39.3 |
| | Female | 108 | 60.7 |
| **Religion** | | | |
| | Muslim | 61 | 34.3 |
| | Orthodox | 77 | 43.3 |
| | Protestant | 31 | 17.4 |
| | Catholic | 6 | 3.4 |
| | Others* | 3 | 1.6 |
| **Ethnicity** | | | |
| | Oromo | 114 | 64 |
| | Amhara | 33 | 18.5 |
| | Sidamo | 17 | 9.6 |
| | Wolayita | 11 | 6.2 |
| | Others** | 3 | 1.7 |
| **Occupation** | | | |
| | Farmer | 46 | 25.8 |
| | House wife | 39 | 21.9 |
| | Merchant | 37 | 20.8 |
| | Daily Laborer | 30 | 19.9 |
| | Government Employee | 15 | 8.4 |
| | Craftsman | 8 | 4.5 |
| | Others*** | 3 | 1.7 |
| **Educational Status** | | | |
| | Illiterate | 60 | 33.7 |
| | Primary School | 71 | 39.9 |
| | Secondary School | 34 | 19.1 |
| | College and Above | 13 | 7.3 |
| **Family Size** | | | |
| | 1–5 members | 126 | 70.8 |
| | 6 and Above Members | 52 | 29.2 |

* Adventists;

** 2 Gurages and 1 kenbata;

*** local alcoholic beverages (Areque).

Note Table 1: indicate the score socio-demographic characteristics towards active trachoma.

respondents stated that they had been dropped out of their school, with 4 (40%t) discontinuing their school between grades 1–4 and the remaining 6 (60%) discontinuing their school between grades 5–8 (Table 4).

## Main proximate variables related to the child and active trachoma

The majority of the faces (56.7%) of the 101 children polled were visible because they were filthy. By just looking at the faces of youngsters, it was discovered that 83 (46.6%) had ocular discharge and 52 (29.2%) had nasal discharge. Furthermore, within 30 seconds of observation, flies were seen on the faces of 135 (75.8%) of the children. Prior to this study, some of

**Table 2. Basic services related to active trachoma of study participants of Arsi Negele town, West Arsi Zone, Oromia, southern Ethiopia, December, 2019.**

| Variable (n = 178) | | Frequency | Percent (100%) |
|---|---|---|---|
| **Number of Rooms** | | | |
| | Less than 2 Rooms | 44 | 24.7 |
| | Exactly 2 Rooms | 92 | 51.7 |
| | Greater than 2 Rooms | 42 | 23.6 |
| **Daily Water Available** | | | |
| | Yes | 56 | 31.5 |
| | No | 122 | 68.5 |
| **Main Source of Water for Domestic Use** | | | |
| | Piped water | 139 | 78.09 |
| | Protected Well | 38 | 21.35 |
| | River | 1 | 0.56 |
| **Travel Time to Fetch Water (two Trips)** | | | |
| | Less than15 minutes | 38 | 21.4 |
| | 16–30 Minutes | 28 | 15.7 |
| | 31–60 Minutes | 18 | 10.1 |
| | Greater than 60 Minutes | 94 | 52.8 |
| **Daily Water consumption of the Family** | | | |
| | Less than 1 Jerrican (20 liters) | 10 | 5.6 |
| | 1–2 Jerrican (20–40 liters) | 46 | 25.84 |
| | 2–3 jerrican (40–60 liters) | 38 | 21.35 |
| | 3–4 Jerrican (60–80 Liters) | 32 | 18 |
| | More than 4 jerrican (80 liters) | 52 | 29.21 |
| **Cooking Condition of the Family** | | | |
| | In the same room with family Lives | 4 | 2.3 |
| | In a Kitchen with living House | 40 | 22.4 |
| | In a kitchen outside the Living House | 130 | 73 |
| | Others* | 4 | 2.3 |
| **The cooking room has window** | | | |
| | Yes | 87 | 48.9 |
| | No | 91 | 51.1 |
| **The cooking room has chimney exhaustion** | | | |
| | Yes | 68 | 38.2 |
| | No | 110 | 61.8 |

* They used the kitchen of their restaurants nearby their house.

Note Table 2: Showed the basic services related to active trachoma of study participants.

the children (69.8%) were not washing their faces or having their family heads wash their faces (Table 5).

## Investigated prevalence of active trachoma

178 Trachoma infections were examined in children aged 1 to 9 years. According to the findings of this study, 39 (21.91%) of all children were identified with active trachoma (TF). Only TF (21.91%) estimated the presence of active trachoma in children [11]. All instances of active trachoma were referred to Shashemene Referral Hospital for treatment by the ocular unit. However, seven of them did not come to the hospital for treatment, and their list, together

**Table 3. Family sanitation, hygiene and awareness about trachoma of the study participants at Arsi Negele Town, West Arsi Zone, Oromia, Southern Ethiopia, December, 2019.**

| Variables (n = 178) | | Frequency | Percent (100%) |
|---|---|---|---|
| **Availability of Family latrine** | | | |
| | Yes | 164 | 92.1 |
| | No | 14 | 7.9 |
| **Availability of Standard Latrine** | | | |
| | Yes | 44 | 24.7 |
| | No | 118 | 66.3 |
| | Not exist or Known | 16 | 9 |
| **Family groups Regularly Use of latrine** | | | |
| | only Adults | 37 | 20.8 |
| | Only Children | 2 | 1.1 |
| | Both Adult and Children | 131 | 73.6 |
| | Not Use at all | 8 | 4.5 |
| **Presence of Feces in the Compound** | | | |
| | Yes | 67 | 37.6 |
| | No | 111 | 62.4 |
| **Management of produced refuses/ garbage** | | | |
| | Burn it | 90 | 50.60 |
| | Burry it | 12 | 6.70 |
| | Dispose in the farm | 32 | 18 |
| | Simply Dispose it in open place | 44 | 24.7 |
| **Availability of domestic Animals** | | | |
| | Yes | 78 | 43.8 |
| | No | 100 | 56.2 |
| **Sleeping Place of Domestic Animals (n = 78)** | | | |
| | In the same room where family lives | 8 | 10.26 |
| | In the same living house but in separate room | 28 | 35.90 |
| | In a shelter separately constructed for them | 41 | 52.56 |
| | Others* | 1 | 1.28 |
| **Awareness about trachoma** | | | |
| | Yes | 115 | 64.6 |
| | No | 63 | 35.4 |

*most of the time sleep in rural area but the cow has come to town for milking only.

Note Table 3: Showed the score of Family Sanitation, Hygiene and Awareness about Trachoma of the study participants.

with the names of their kebeles, was submitted to the Arsi Negele town health office for health extension workers to trace and follow them (Fig 1). Regarding frequency of face washing, 19 (17.43%) children washed their faces once per day, 48 (44.04%) children washed their faces two or more times per day and 42 (38.53%) stay unwashed for longer than a week (Fig 2).

## Variables associated with the prevalence of active trachoma

Bivariate logistic regression was used to select the association between the presence of active trachoma and risk factors. Thus, only nine variables or factors out of the total number of factors surveyed showed an association (P< 0.25 and 95% CI) with the prevalence of active trachoma. All significant variables (P< 0.25) were then included in a multivariate logistic

**Table 4. Basic characteristics of studied children of Arsi Negele Town, West Arsi Zone, Oromia, Southern Ethiopia, and December, 2019.**

| Variables (n = 178) | | Frequency | Percent (100%) |
|---|---|---|---|
| **Age of children in years** | | | |
| | 1–5 years | 128 | 71.9 |
| | 6–9 years | 50 | 28.1 |
| **Sex of children** | | | |
| | Male | 93 | 52.2 |
| | Female | 85 | 47.8 |
| **Educational Status of Children** | | | |
| | Too young to go school (preschool) | 77 | 43.3 |
| | Illiterate | 28 | 15.7 |
| | Dropped out of school (Discontinued) | 10 | 5.6 |
| | Attending School | 63 | 35.4 |
| **Highest grade completed (School dropped; n = 10)** | | | |
| | Grade 1–4 | 4 | 40 |
| | Grade 5–8 | 6 | 60 |

regression using a forward LR stepwise strategy (because it predicted active trachoma to the greatest extent (83%t) of all other approaches), with confounding factors adjusted at P< 0.05 and CI 95%.

The majorly associated factors identified by the results of this study were flies observed on children's faces (AOR = 3.421; 95%, CI: (1.432–8.171)), unclean faces of children (AOR = 3.99; 95%, CI: (1.427–11.158)), children's face washing habits (AOR = 3.064; 95%, CI: (1.273–7.373)), and not using soap when washing children's faces (AOR = 4.564; 95%, CI: (1.561–13.156)) of surveyed children aged 1–9 years at Arsi Negele town.

**Table 5. Main proximate variables related to active trachoma among studied children of Arsi Negele Town, West Arsi Zone, Oromia, Southern Ethiopia, December, 2019.**

| Variable | | Frequency (n = 178) | Percent (100%) |
|---|---|---|---|
| **Cleanliness of child faces** | | | |
| | Clean | 77 | 43.3 |
| | Not clean | 101 | 56.7 |
| **Ocular discharge** | | | |
| | Yes | 83 | 46.6 |
| | No | 95 | 53.4 |
| **Nasal discharge** | | | |
| | Yes | 52 | 29.2 |
| | No | 126 | 70.8 |
| **Fly observed on the child face** | | | |
| | Yes | 135 | 75.8 |
| | No | 43 | 24.2 |
| **The child washes his or her face** | | | |
| | Yes | 109 | 61.2 |
| | No | 69 | 38.8 |
| **Child use soap when washing face** | | | |
| | Yes | 70 | 39.33 |
| | Yes, some times | 39 | 21.91 |
| | No | 69 | 38.76 |

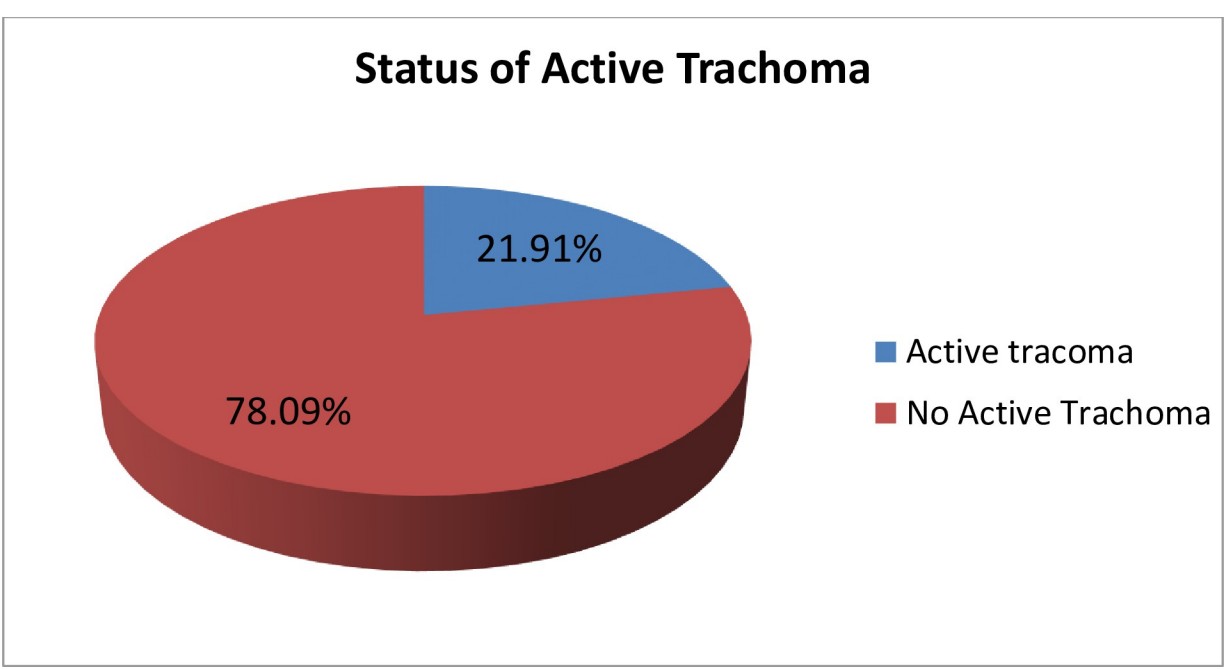

**Fig 1. Status of active trachoma among children aged 1–9 years in Arsi Negele town.**

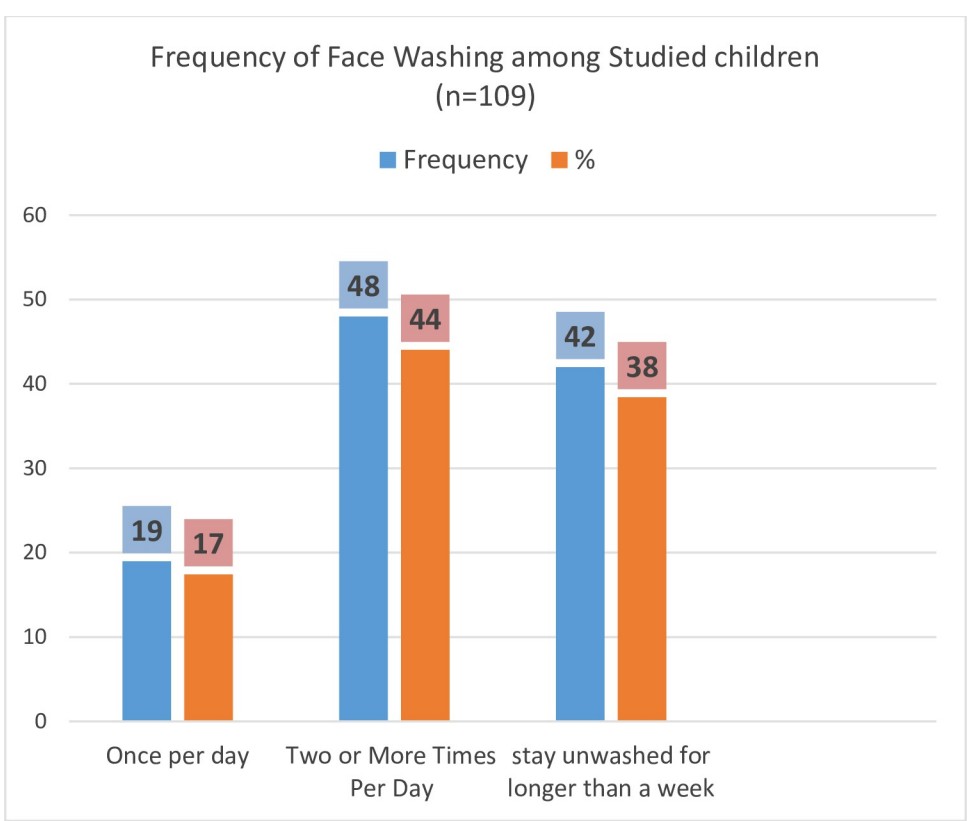

**Fig 2. The frequency of face washing among children aged 1–9 years in Arsi Negele town.**

Thus, flies on children's faces, cleanliness of children's faces, children's face washing habits, and soap use when cleaning faces were discovered to be independent predictors or related factors for the prevalence of active trachoma among children aged 1–9 years. at Arsi Negele town (Table 6).

## Discussion

The aim of this study was to determine the current prevalence of active trachoma and its associated factors in children aged 1 to 9 years in Arsi Negele Town. According to the findings of this study, the prevalence of active trachoma among children aged 1–9 years was 21.91%, with the majority of cases being TF, which is higher than the WHO recommendation for trachoma elimination (5% of active trachoma).

**Table 6. Variables associated with active trachoma among children aged 1–9 years at Arsi Negele Town, West Arsi, Oromia, Southern Ethiopia, December 2019.**

| | | Associated Variables with Active Trachoma | | | |
|---|---|---|---|---|---|
| | **Variables** | **Active Trachoma** | | **COR,95% CI** | **AOR,95%CI** |
| | | Yes | No | | |
| **Trachoma Awareness** | | | | | |
| | Yes | 18 | 97 | 1 | |
| | No | 21 | 42 | 0.371 (0.180–0.767) | |
| **Blindness occurred** | | | | | |
| | Yes | 14 | 82 | 1 | |
| | No | 5 | 12 | 0.384 (0.177–0.833) | |
| | I do not Know | 20 | 45 | 0.937 (0.291–3.017) | |
| **Cleanliness of child faces** | | | | | |
| | Clean | 5 | 72 | 1 | |
| | Not Clean | 34 | 67 | 0.174(0.069–0.442) | 3.99(1.427–11.158)* |
| **Ocular Discharge** | | | | | |
| | Yes | 25 | 58 | 2.494(1.195–5.206) | |
| | No | 14 | 81 | 1 | |
| **Nasal Discharge** | | | | | |
| | Yes | 21 | 31 | 4.065(1.929–8.566) | |
| | No | 18 | 108 | 1 | |
| **Fly Observed on Child face** | | | | | |
| | Yes | 28 | 44 | 5.496(2.510–12.033) | 3.427(1.432–8.171)* |
| | No | 11 | 95 | 1 | |
| **Face washing Habits** | | | | | |
| | Yes | 13 | 96 | 1 | |
| | No | 26 | 43 | 0.224(0.105–0.477) | 3.064 (1.273–7.373)* |
| **Frequency of face washing** | | | | | |
| | Once per day | 8 | 30 | 1 | |
| | Two or more times per day | 6 | 69 | 0.427(0.169–1.077) | |
| | unwashed for longer a week | 25 | 40 | 0.139(0.053–0.368) | |
| **Children uses soap** | | | | | |
| | Yes | 6 | 64 | 1 | |
| | Yes, some times | 22 | 51 | 0.205(0.06–0.614) | |
| | No | 11 | 24 | 0.941(0.394–2.249) | 4.564 (1.561–13.342)* |

NB. 1 = Reference category; * P-value 0.05–0.01; ** 0.01–0.001; *** p value < 0.001.

This finding is consistent with studies conducted in similar areas, such as Maksegnit Town and Dessie Zuria in the Amhara region, where rates were 23.8% and 21.6%, respectively [13]. The odd ratio of flies detected on the faces of children in this study suggested that the probabilities of having active trachoma among children with flies noticed on their faces were three times higher than those without flies observed on their faces.

Previous research found that flies on the face were a substantial risk factor for active trachoma, with children with flies on their faces more than twice as likely to have active trachoma as children without flies on their faces. According to a Nigerian study, a youngster with flies on his or her face is four times more likely to have active trachoma than a child without insects [14].

However, some members of the community do not associate flies on the face with trachoma illness [15].

This study also found that children with filthy faces were three times more likely to have active trachoma than children with clean faces at the time of observation. In line with this, numerous investigations have proven that unclean faces are a key source of active trachoma and are significantly associated with a higher prevalence of the disease. Furthermore, ocular or nasal discharge might cause a dirty face, which has been associated to active trachoma in several settings [16].

This study also discovered an unusual ratio of children's face-washing habits, which could be interpreted as the odds of having active trachoma among children who did not have face-washing habits in their previous life being three times higher than those children who did have face-washing habits in their previous life. In line with this finding, a study in Gondar, Zuria, revealed that children's face-washing practices were substantial predictors of active trachoma [17].

In the current study, the odd ratio of using soap while washing children's faces can be expressed as the odds of having active trachoma among children who did not use soap while washing their faces being more than four times greater than those who did use soap while washing their faces being more than four times greater than those who did use soap while washing their faces. Research conducted in Ethiopia discovered that children who did not use soap to wash their faces were more than twice as likely to acquire active trachoma as those who claimed to use soap [18].

This study's odds ratio for soap consumption during face washing is higher than the relevant study stated above. This could be because the population is growing and trachoma is more contagious in populated areas.

## Limitation of study

The mothers couldn't remember the exact ages of their children. There could be recall biases.

## Conclusion & recommendation

The prevalence of active trachoma among children aged 1–9 years in Arsi Negele town was found to be greater than the WHO suggested criteria (5% prevalence) to commence trachoma control recommendations, indicating that active trachoma remains a public health problem.in the Arsi Negele Town. The key associated factors were flies observed on children's faces, filthy children's faces, poor face washing practices, and the lack of use of soap while washing children's faces, all of which require attention in order to work effectively on trachoma prevention as a public health problem.

As a result, it is recommended that coordinated work on implementing the WHO-endorsed SAFE strategy in particular, and improving the overall living conditions of the community in

general, be carried out by a collaborative effort of the West Arsi Zonal Health Office, Arsi Negele Town Health Office, Arsi Negele Town Water and Sewerage Authority, FHF-E Shashemene Cluster office, One-WaSH sector, and other concerned partners or potential implementers.

## Supporting information

**S1 Dataset.**
(XLSX)

## Author Contributions

**Conceptualization:** Jemal Mekonnen, Jeylan Kassim.

**Data curation:** Jemal Mekonnen, Jeylan Kassim.

**Formal analysis:** Jemal Mekonnen, Jeylan Kassim, Muluneh Ahmed.

**Funding acquisition:** Jemal Mekonnen, Jeylan Kassim.

**Investigation:** Jemal Mekonnen, Jeylan Kassim, Muluneh Ahmed, Negeso Gebeyehu.

**Methodology:** Jemal Mekonnen, Jeylan Kassim.

**Project administration:** Jemal Mekonnen, Jeylan Kassim, Muluneh Ahmed.

**Resources:** Jemal Mekonnen.

**Software:** Jemal Mekonnen, Jeylan Kassim, Muluneh Ahmed, Negeso Gebeyehu.

**Supervision:** Jeylan Kassim.

**Validation:** Jemal Mekonnen, Jeylan Kassim, Muluneh Ahmed, Negeso Gebeyehu.

**Visualization:** Jemal Mekonnen, Jeylan Kassim, Muluneh Ahmed, Negeso Gebeyehu.

**Writing – original draft:** Jemal Mekonnen, Muluneh Ahmed.

**Writing – review & editing:** Jemal Mekonnen, Muluneh Ahmed, Negeso Gebeyehu.

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
