## [Decision Letter · Decision Letter 0]

21 Mar 2022

PONE-D-21-40184Prevalence of active trachoma and associated factors among children 1-9 years old at Arsi Negele Town , West Arsi Zone, Oromia Regional State, Southern EthiopiaPLOS ONE

Dear Dr. M. Ahmed-Endris,

Thank you for submitting your manuscript to PLOS ONE. After careful consideration, we feel that it has merit but does not fully meet PLOS ONE’s publication criteria as it currently stands. Therefore, we invite you to submit a revised version of the manuscript that addresses the points raised during the review process.

We look forward to receiving your revised manuscript.

Kind regards,

Hans-Uwe Dahms, Ph.D.

Academic Editor

PLOS ONE

Journal Requirements:

2. We note that you have referenced (ie. Bewick et al. [5]) which has currently not yet been accepted for publication. Please remove this from your References and amend this to state in the body of your manuscript: (ie “Bewick et al. [Unpublished]”) as detailed online in our guide for authors

Additional Editor Comments (if provided):

This manuscript focussed to assess the factor associated with active trachoma among the children with age group of 1- 9 years old. The prevalence of active trachoma was determined with 21.91% among the screened 178 children. The children taken for the study were categorized for different level like children have clean faces, unclean faces, face washing habitats, not using soap while face and these factors were found as statistically significance which was associated with the prevalence of active trachoma. The high prevalence of active trachoma was found more association with the factors of face washing practices and the lack of soap use while washing faces. Filthy faces and flies on the children faces were observed as more risk factors to have active trachoma.

Queries;

1. The study was conducted between 1-9 years old children but more numbers (71.9%) were present between 1-5 years than 6-9 years (28.1%) in Arsi Negele Town also in line 353 they have motioned that is higher than the WHO recommendation for trachoma elimination. The age associated factor for this trachoma infection is lacking in the discussion part and only the result is provided. The authors can modify it accordingly

2. Since it’s a medical terminology the abbreviation for TF (Trachoma Folliculitis) can be produced in the text itself particularly in the abstract and I think no need to give separate list for this especially in abbreviation. Sudden confusion may rise in the abstract while reading the terminology TF.

3. The regional wise information regard the hygienic practices of the localized study area was given as nice but relating with the hygienic practices of environmental (regional) factors associated with trachoma infection is not scientifically discussed in this manuscript.

4. However the clean and unclean faces, flies over the faces were observed in this study as the significant factors associated with the Trachoma infection, but the condition caused blindness and which stage has more risk towards it is essentially need to illustrate in this study.

5. Lacking of potential figures in this manuscript.

6. Over all the manuscript is looks like a report of the clinical study but the scientific illustration from the observation is not fulfilled the criteria and does not give a representation of scientific research temper.

Hence I request the authors to complete and rework with the above suggestions under major revision of the manuscript.(less...)

Reviewers' comments:

Reviewer's Responses to Questions

**Comments to the Author**

1. Is the manuscript technically sound, and do the data support the conclusions?

Reviewer #1: No

Reviewer #2: Yes

2. Has the statistical analysis been performed appropriately and rigorously? 

Reviewer #1: Yes

Reviewer #2: Yes

3. Have the authors made all data underlying the findings in their manuscript fully available?

Reviewer #1: No

Reviewer #2: Yes

4. Is the manuscript presented in an intelligible fashion and written in standard English?

Reviewer #1: Yes

Reviewer #2: Yes

5. Review Comments to the Author

Reviewer #1: TRACHOMA – A PUBLIC HEALTH ISSUE

To live a healthy lifestyle we need to follow a proper healthy diet and follow proper sanitation and stay hygiene, otherwise we will face a variety of diseases and allergies which could lead to severe medical issues and have to follow a lot of medications. So, the paper which im reviewing today is a topic related to the improper sanitation issues which cause a blindness disorder called Trachoma, which is commonly found in the areas with improper hygiene and sanitation mostly such kind of disorder is found in African countries like Nigeria ,Sudan and many other sub Saharan African nations.

Trachoma is a kind of disease which causes conjunctival infection in eye and eventually cause infection in retina and hence cause blindness. The infection spreads easily from person to person with nasal and ocular secretions, and along with sharing of towels, hankerchief etc. This type of infection spreads due to lack of proper sanitation and hygiene. It is mostly seen in the African children aged 1-9 years of age. A survey was considered among the children of aged 1-9 living in dirty quarters and dirty conditions like no proper sanitation and the areas without good water facilities and also the children living in neat and clean hygienic environment. A team has been formed to conduct the survey and they collected the data of their survey and made a list of the population and checked the impact of the infection by using statistical analysis by the help of IBM IT industry. The list was made by focusing on the age, sex , ethnicity like indirect factors and direct factors like water and cooking conditions which play a vital role to check the spreading and virulence of the pathogen.

After collecting the data they found that the infection spread faster to the children staying in unhygienic conditions rather those children staying in hygienic environment. The unhygienic environment gives exposures to children towards conditions like faeces in their house compound, using of dirty towels, flies moving around the faces of the children, improper washing of face without using soap and without using hand washes sanitizer etc, lack of water supply and using of dirty contaminated water of rivers and ponds. The infection is basically a bacterial infection caused by Chlamydia trachomatis a bacteria which causes conjunctival infection initially and eventually cause retinal damage. The symptoms of Trachoma is irritation in eyes, watery eyes, redness, improper vision etc.

According to WHO it says 5% prevalence to tackle Trachoma but in reality after doing a legitimate survey they found that the prevalence of active Trachoma in children 1-9 was about 21.91%. The Bacteria is present on the contaminated surfaces and unhygienic locations years and onto which flies used to fly around and these flies carries the pathogen and contaminate the food, household things, skin surface etc and hence cause the spreading of the infection. The study also found that the children with the filthy faces were three times more likely to have active Trachoma.

So, according to this paper it says by taking proper sanitation and maintaining well sanitized environment and good living conditions can eradicate this microbial pathogen which is really a good report made by the people involved in it. I really appreciate their work to make us aware regarding this serious infection, But as a reviewer of this paper I would suggest if they would have mention the virulence of this particular pathogen causing conjunctival infection and its complete pathogenic period in the child’s body in a molecular biology and biochemistry context like how it eventually leads to retinal damage then it would be much more interesting and informative. And also I got a question regarding how the bacteria isn’t get damaged after entering the eye because as we known that eye contains lysosomal enzyme which has the capability to destroy bacterial cell wall. These information should be present in this paper. However it’s really a good survey……….

Reviewer #2: This manuscript focussed to assess the factor associated with active trachoma among the children with age group of 1- 9 years old. The prevalence of active trachoma was determined with 21.91% among the screened 178 children. The children taken for the study were categorized for different level like children have clean faces, unclean faces, face washing habitats, not using soap while face and these factors were found as statistically significance which was associated with the prevalence of active trachoma. The high prevalence of active trachoma was found more association with the factors of face washing practices and the lack of soap use while washing faces. Filthy faces and flies on the children faces were observed as more risk factors to have active trachoma.

Queries;

1. The study was conducted between 1-9 years old children but more numbers (71.9%) were present between 1-5 years than 6-9 years (28.1%) in Arsi Negele Town also in line 353 they have motioned that is higher than the WHO recommendation for trachoma elimination. The age associated factor for this trachoma infection is lacking in the discussion part and only the result is provided. The authors can modify it accordingly

2. Since it’s a medical terminology the abbreviation for TF (Trachoma Folliculitis) can be produced in the text itself particularly in the abstract and I think no need to give separate list for this especially in abbreviation. Sudden confusion may rise in the abstract while reading the terminology TF.

3. The regional wise information regard the hygienic practices of the localized study area was given as nice but relating with the hygienic practices of environmental (regional) factors associated with trachoma infection is not scientifically discussed in this manuscript.

4. However the clean and unclean faces, flies over the faces were observed in this study as the significant factors associated with the Trachoma infection, but the condition caused blindness and which stage has more risk towards it is essentially need to illustrate in this study.

5. Lacking of potential figures in this manuscript.

6. Over all the manuscript is looks like a report of the clinical study but the scientific illustration from the observation is not fulfilled the criteria and does not give a representation of scientific research temper.

Hence I request the authors to complete and rework with the above suggestions under major revision of the manuscript.

6. PLOS authors have the option to publish the peer review history of their article (what does this mean?). If published, this will include your full peer review and any attached files.

Reviewer #1: No

Reviewer #2: No

---

## [Author Response · Author response to Decision Letter 0]

7 Aug 2022

we have modified the manuscript as per comment given. we kindly request the editors and reviewers to re evaluate it again

---

## [Editor Report · Decision Letter 1]

16 Aug 2022

Prevalence of active trachoma and associated factors among children 1-9 years old at Arsi Negele Town , West Arsi Zone, Oromia Regional State, Southern Ethiopia

PONE-D-21-40184R1

Dear Dr. Endris,

We’re pleased to inform you that your manuscript has been judged scientifically suitable for publication and will be formally accepted for publication once it meets all outstanding technical requirements.

Kind regards,

Hans-Uwe Dahms, Ph.D.

Academic Editor

PLOS ONE

Additional Editor Comments (optional):

This MS has now made substantial advances and is accepted for publication.

I trust that formal in house corrections will still be made.
---

## [Editor Report · Acceptance letter]

27 Sep 2022

PONE-D-21-40184R1 

Prevalence of active trachoma and associated factors among children 1-9 years old at Arsi Negele Town , West Arsi Zone, Oromia Regional State, Southern Ethiopia 

Dear Dr. Endris:

I'm pleased to inform you that your manuscript has been deemed suitable for publication in PLOS ONE. Congratulations! Your manuscript is now with our production department. 

Kind regards, 

on behalf of

Dr. Hans-Uwe Dahms 

Academic Editor

PLOS ONE